# Generalized Method-of-Moments for Rank Aggregation

**Hossein Azari Soufiani**
SEAS
Harvard University
azari@fas.harvard.edu

**William Z. Chen**
Statistics Department
Harvard University
wchen@college.harvard.edu

**David C. Parkes**
SEAS
Harvard University
parkes@eecs.harvard.edu

**Lirong Xia**
Computer Science Department
Rensselaer Polytechnic Institute
Troy, NY 12180, USA
xial@cs.rpi.edu

## Abstract

In this paper we propose a class of efficient Generalized Method-of-Moments (GMM) algorithms for computing parameters of the Plackett-Luce model, where the data consists of full rankings over alternatives. Our technique is based on breaking the full rankings into pairwise comparisons, and then computing parameters that satisfy a set of generalized moment conditions. We identify conditions for the output of GMM to be unique, and identify a general class of consistent and inconsistent breakings. We then show by theory and experiments that our algorithms run significantly faster than the classical Minorize-Maximization (MM) algorithm, while achieving competitive statistical efficiency.

## 1  Introduction

In many applications, we need to aggregate the preferences of agents over a set of alternatives to produce a joint ranking. For example, in systems for ranking the quality of products, restaurants, or other services, we can generate an aggregate rank through feedback from individual users. This idea of *rank aggregation* also plays an important role in multiagent systems, meta-search engines [4], belief merging [5], crowdsourcing [15], and many other e-commerce applications.

A standard approach towards rank aggregation is to treat input rankings as data generated from a probabilistic model, and then learn the MLE of the input data. This idea has been explored in both the machine learning community and the (computational) social choice community. The most popular statistical models are the Bradley-Terry-Luce model (BTL for short) [2, 13], the Plackett-Luce model (PL for short) [17, 13], the random utility model [18], and the Mallows (Condorcet) model [14, 3]. In machine learning, researchers have focused on designing efficient algorithms to estimate parameters for popular models; e.g. [8, 12, 1]. This line of research is sometimes referred to as *learning to rank* [11].

Recently, Negahban et al. [16] proposed a rank aggregation algorithm, called *Rank Centrality* (RC), based on computing the stationary distribution of a Markov chain whose transition matrix is defined according to the data (pairwise comparisons among alternatives). The authors describe the approach as being model independent, and prove that for data generated according to BTL, the output of RC converges to the ground truth, and the performance of RC is almost identical to the performance of

MLE for BTL. Moreover, they characterized the convergence rate and showed experimental comparisons.

**Our Contributions.** In this paper, we take a *generalized method-of-moments (GMM)* point of view towards rank aggregation. We first reveal a new and natural connection between the RC algorithm [16] and the BTL model by showing that RC algorithm can be interpreted as a GMM estimator applied to the BTL model.

The main technical contribution of this paper is a class of GMMs for parameter estimation under the PL model, which generalizes BTL and the input consists of full rankings instead of pairwise comparisons as in the case of BTL and RC algorithm.

Our algorithms first *break* full rankings into pairwise comparisons, and then solve the generalized moment conditions to find the parameters. Each of our GMMs is characterized by a way of breaking full rankings. We characterize conditions for the output of the algorithm to be unique, and we also obtain some general characterizations that help us to determine which method of breaking leads to a consistent GMM. Specifically, *full breaking* (which uses all pairwise comparisons in the ranking) is consistent, but *adjacent breaking* (which only uses pairwise comparisons in adjacent positions) is inconsistent.

We characterize the computational complexity of our GMMs, and show that the asymptotic complexity is better than for the classical Minorize-Maximization (MM) algorithm for PL [8]. We also compare statistical efficiency and running time of these methods experimentally using both synthetic and real-world data, showing that all GMMs run much faster than the MM algorithm.

For the synthetic data, we observe that many consistent GMMs converge as fast as the MM algorithm, while there exists a clear tradeoff between computational complexity and statistical efficiency among consistent GMMs.

Technically our technique is related to the random walk approach [16]. However, we note that our algorithms aggregate *full rankings* under PL, while the RC algorithm aggregates *pairwise comparisons*. Therefore, it is quite hard to directly compare our GMMs and RC fairly since they are designed for different types of data. Moreover, by taking a GMM point of view, we prove the consistency of our algorithms on top of theories for GMMs, while Negahban et al. proved the consistency of RC directly.

## 2 Preliminaries

Let $\mathcal{C} = \{c_1, .., c_m\}$ denote the set of $m$ alternatives. Let $D = \{d_1, \ldots, d_n\}$ denote the data, where each $d_j$ is a full ranking over $\mathcal{C}$. The PL model is a parametric model where each alternative $c_i$ is parameterized by $\gamma_i \in (0, 1)$, such that $\sum_{i=1}^m \gamma_i = 1$. Let $\vec{\gamma} = (\gamma_1, \ldots, \gamma_m)$ and $\Omega$ denote the parameter space. Let $\bar{\Omega}$ denote the closure of $\Omega$. That is, $\bar{\Omega} = \{\vec{\gamma} : \forall i, \gamma_i \geq 0 \text{ and } \sum_{i=1}^m \gamma_i = 1\}$. Given $\vec{\gamma}^* \in \Omega$, the probability for a ranking $d = [c_{i_1} \succ c_{i_2} \succ \cdots \succ c_{i_m}]$ is defined as follows.

$$\mathrm{Pr}_{\mathrm{PL}}(d|\vec{\gamma}) = \frac{\gamma_{i_1}}{\sum_{l=1}^m \gamma_{i_l}} \times \frac{\gamma_{i_2}}{\sum_{l=2}^m \gamma_{i_l}} \times \cdots \times \frac{\gamma_{i_{m-1}}}{\gamma_{i_{m-1}} + \gamma_{i_m}}$$

In the BTL model, the data is composed of pairwise comparisons instead of rankings, and the model is parameterized in the same way as PL, such that $\mathrm{Pr}_{\mathrm{BTL}}(c_{i_1} \succ c_{i_2}|\vec{\gamma}) = \frac{\gamma_{i_1}}{\gamma_{i_1} + \gamma_{i_2}}$. BTL can be thought of as a special case of PL via marginalization, since $\mathrm{Pr}_{\mathrm{BTL}}(c_{i_1} \succ c_{i_2}|\vec{\gamma}) = \sum_{d:c_{i_1} \succ c_{c_2}} \mathrm{Pr}_{\mathrm{PL}}(d|\vec{\gamma})$. In the rest of the paper, we denote $\mathrm{Pr} = \mathrm{Pr}_{\mathrm{PL}}$.

*Generalized Method-of-Moments (GMM)* provides a wide class of algorithms for parameter estimation. In GMM, we are given a parametric model whose parametric space is $\Omega \subseteq \mathbb{R}^m$, an infinite series of $q \times q$ matrices $\mathcal{W} = \{W_t : t \geq 1\}$, and a column-vector-valued function $g(d, \vec{\gamma}) \in \mathbb{R}^q$. For any vector $\vec{a} \in \mathbb{R}^q$ and any $q \times q$ matrix $W$, we let $\|\vec{a}\|_W = (\vec{a})^T W \vec{a}$. For any data $D$, let $g(D, \vec{\gamma}) = \frac{1}{n} \sum_{d \in D} g(d, \vec{\gamma})$, and the GMM method computes parameters $\vec{\gamma}' \in \Omega$ that minimize $\|g(D, \vec{\gamma}')\|_{W_n}$, formally defined as follows:

$$\mathrm{GMM}_g(D, \mathcal{W}) = \{\vec{\gamma}' \in \Omega : \|g(D, \vec{\gamma}')\|_{W_n} = \inf_{\vec{\gamma} \in \Omega} \|g(D, \vec{\gamma})\|_{W_n}\} \qquad (1)$$

Since $\Omega$ may not be compact (as is the case for PL), the set of parameters $\mathrm{GMM}_g(D, \mathcal{W})$ can be empty. A GMM is *consistent* if and only if for any $\vec{\gamma}^* \in \Omega$, $\mathrm{GMM}_g(D, \mathcal{W})$ converges in probability to $\vec{\gamma}^*$ as $n \to \infty$ and the data is drawn i.i.d. given $\vec{\gamma}^*$. Consistency is a desirable property for GMMs.

It is well-known that $\text{GMM}_g(D, \mathcal{W})$ is consistent if it satisfies some regularity conditions plus the following condition [7]:

**Condition 1.** $E_{d|\vec{\gamma}^*}[g(d, \vec{\gamma})] = 0$ if and only if $\vec{\gamma} = \vec{\gamma}^*$.

**Example 1. MLE as a consistent GMM:** *Suppose the likelihood function is twice-differentiable, then the MLE is a consistent GMM where $g(d, \vec{\gamma}) = \bigtriangledown_{\vec{\gamma}} \log \Pr(d|\vec{\gamma})$ and $W_n = I$.*

**Example 2.** *Negahban et al. [16] proposed the* Rank Centrality (RC) *algorithm that aggregates pairwise comparisons $D_P = \{Y_1, \ldots, Y_n\}$.[1] Let $a_{ij}$ denote the number of $c_i \succ c_j$ in $D_P$ and it is assumed that for any $i \neq j$, $a_{ij} + a_{ji} = k$. Let $d_{max}$ denote the maximum pairwise defeats for an alternative. RC first computes the following $m \times m$ column stochastic matrix:*

$$P_{RC}(D_P)_{ij} = \left\{ \begin{array}{rl} a_{ij}/(kd_{max}) & \text{if } i \neq j \\ 1 - \sum_{l \neq i} a_{li}/(kd_{max}) & \text{if } i = j \end{array} \right.$$

*Then, RC computes $(P_{RC}(D_P))^T$'s stationary distribution $\vec{\gamma}$ as the output.*

*Let $X^{c_i \succ c_j}(Y) = \left\{ \begin{array}{ll} 1 & \text{if } Y = [c_i \succ c_j] \\ 0 & \text{otherwise} \end{array} \right.$ and $P_{RC}^*(Y) = \left\{ \begin{array}{rl} X^{c_i \succ c_j} & \text{if } i \neq j \\ -\sum_{l \neq i} X^{c_l \succ c_i} & \text{if } i = j \end{array} \right.$.*

*Let $g_{RC}(d, \vec{\gamma}) = P_{RC}^*(d) \cdot \vec{\gamma}$. It is not hard to check that the output of RC is the output of $\text{GMM}_{g_{RC}}$. Moreover, $\text{GMM}_{g_{RC}}$ satisfies Condition 1 under the BTL model, and as we will show later in Corollary 4, $\text{GMM}_{g_{RC}}$ is consistent for BTL.*

## 3 Generalized Method-of-Moments for the Plakett-Luce model

In this section we introduce our GMMs for rank aggregation under PL. In our methods, $q = m$, $W_n = I$ and $g$ is linear in $\vec{\gamma}$. We start with a simple special case to illustrate the idea.

**Example 3.** *For any full ranking $d$ over $\mathcal{C}$, we let*

- $X^{c_i \succ c_j}(d) = \left\{ \begin{array}{ll} 1 & c_i \succ_d c_j \\ 0 & \text{otherwise} \end{array} \right.$

- $P(d)$ *be an $m \times m$ matrix where* $P(d)_{ij} = \left\{ \begin{array}{rl} X^{c_i \succ c_j}(d) & \text{if } i \neq j \\ -\sum_{l \neq i} X^{c_l \succ c_i}(d) & \text{if } i = j \end{array} \right.$

- $g_F(d, \vec{\gamma}) = P(d) \cdot \vec{\gamma}$ *and* $P(D) = \frac{1}{n} \sum_{d \in D} P(d)$

*For example, let $m = 3$, $D = \{[c_1 \succ c_2 \succ c_3], [c_2 \succ c_3 \succ c_1]\}$. Then $P(D) = \left[ \begin{array}{ccc} -1 & 1/2 & 1/2 \\ 1/2 & -1/2 & 1 \\ 1/2 & 0 & -3/2 \end{array} \right]$. The corresponding GMM seeks to minimize $\|P(D) \cdot \vec{\gamma}\|_2^2$ for $\vec{\gamma} \in \Omega$.*

*It is not hard to verify that* $(E_{d|\vec{\gamma}^*}[P(d)])_{ij} = \left\{ \begin{array}{rl} \frac{\gamma_i^*}{\gamma_i^* + \gamma_j^*} & \text{if } i \neq j \\ -\sum_{l \neq i} \frac{\gamma_i^*}{\gamma_i^* + \gamma_l^*} & \text{if } i = j \end{array} \right.$, *which means that*

$E_{d|\vec{\gamma}^*}[g_F(d, \vec{\gamma}^*)] = E_{d|\vec{\gamma}^*}[P(d)] \cdot \vec{\gamma}^* = 0$. *It is not hard to verify that $\vec{\gamma}^*$ is the only solution to $E_{d|\vec{\gamma}^*}[g_F(d, \vec{\gamma})] = 0$. Therefore, $\text{GMM}_{g_F}$ satisfies Condition 1. Moreover, we will show in Corollary 3 that $\text{GMM}_{g_F}$ is consistent for PL.*

In the above example, we count all pairwise comparisons in a full ranking $d$ to build $P(d)$, and define $g = P(D) \cdot \vec{\gamma}$ to be linear in $\vec{\gamma}$. In general, we may consider some subset of pairwise comparisons. This leads to the definition of our class of GMMs based on the notion of *breakings*. Intuitively, a breaking is an undirected graph over the $m$ positions in a ranking, such that for any full ranking $d$, the pairwise comparisons between alternatives in the $i$th position and $j$th position are counted to construct $P_G(d)$ if and only if $\{i, j\} \in G$.

**Definition 1.** *A breaking is a non-empty undirected graph $G$ whose vertices are $\{1, \ldots, m\}$. Given any breaking $G$, any full ranking $d$ over $\mathcal{C}$, and any $c_i, c_j \in \mathcal{C}$, we let*

- $X_G^{c_i \succ c_j}(d) = \begin{cases} 1 & \{\mathrm{Pos}(c_i,d), \mathrm{Pos}(c_j,d)\} \in G \text{ and } c_i \succ_d c_j \\ 0 & otherwise \end{cases}$, where $\mathrm{Pos}(c_i,d)$ is the position of $c_i$ in $d$.

- $P_G(d)$ be an $m \times m$ matrix where $P_G(d)_{ij} = \begin{cases} X_G^{c_i \succ c_j}(d) & if\ i \neq j \\ -\sum_{l \neq i} X_G^{c_l \succ c_i}(d) & if\ i = j \end{cases}$

- $g_G(d, \vec{\gamma}) = P_G(d) \cdot \vec{\gamma}$

- $GMM_G(D)$ be the GMM method that solves Equation (1) for $g_G$ and $W_n = I$.[2]

In this paper, we focus on the following breakings, illustrated in Figure 1.

- **Full breaking**: $G_F$ is the complete graph. Example 3 is the GMM with full breaking.

- **Top-$k$ breaking**: for any $k \leq m$, $G_T^k = \{\{i,j\} : i \leq k, j \neq i\}$.

- **Bottom-$k$ breaking**: for any $k \geq 2$, $G_B^k = \{\{i,j\} : i,j \geq m+1-k, j \neq i\}$.[3]

- **Adjacent breaking**: $G_A = \{\{1,2\}, \{2,3\}, \ldots, \{m-1,m\}\}$.

- **Position-$k$ breaking**: for any $k \geq 2$, $G_P^k = \{\{k,i\} : i \neq k\}$.

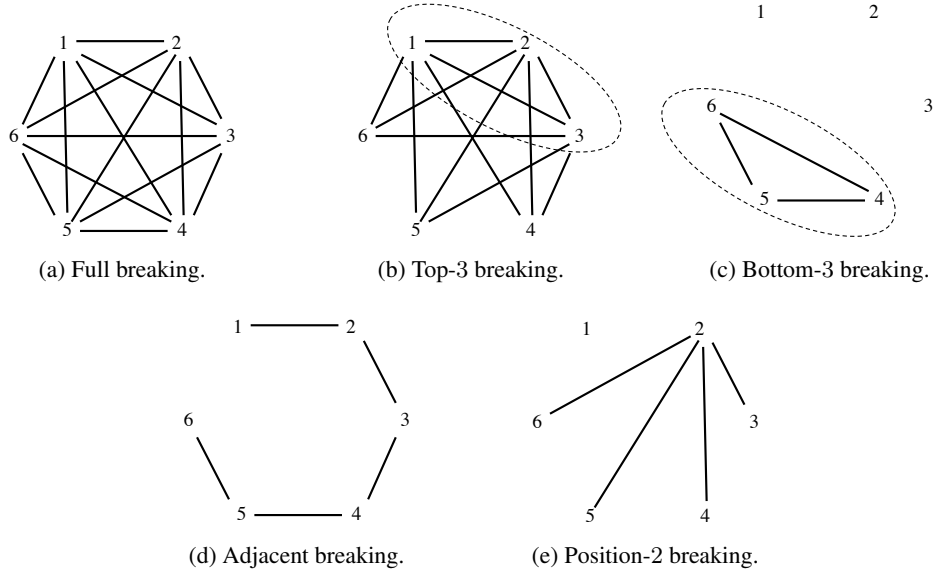

(a) Full breaking.     (b) Top-3 breaking.     (c) Bottom-3 breaking.

(d) Adjacent breaking.     (e) Position-2 breaking.

Figure 1: Example breakings for $m = 6$.

Intuitively, the full breaking contains all the pairwise comparisons that can be extracted from each agent's full rank information in the ranking; the top-$k$ breaking contains all pairwise comparisons that can be extracted from the rank provided by an agent when she only reveals her top $k$ alternatives and the ranking among them; the bottom-$k$ breaking can be computed when an agent only reveals her bottom $k$ alternatives and the ranking among them; and the position-$k$ breaking can be computed when the agent only reveals the alternative that is ranked at the $k$th position and the set of alternatives ranked in lower positions.

We note that $G_T^m = G_B^m = G_F$, $G_T^1 = G_P^1$, and for any $k \leq m - 1$, $G_T^k \cup G_B^{m-k} = G_F$, and $G_T^k = \bigcup_{l=1}^{k} G_P^l$.

We are now ready to present our GMM algorithm (Algorithm 1) parameterized by a breaking $G$.

**Algorithm 1:** $\text{GMM}_G(D)$
---
**Input**: A breaking $G$ and data $D = \{d_1, \ldots, d_n\}$ composed of full rankings.
**Output**: Estimation $\text{GMM}_G(D)$ of parameters under PL.
  **1** Compute $P_G(D) = \frac{1}{n} \sum_{d \in D} P_G(d)$ in Definition 1.
  **2** Compute $\text{GMM}_G(D)$ according to (1).
  **3 return** $\text{GMM}_G(D)$.
---

Step 2 can be further simplified according to the following theorem. Due to the space constraints, most proofs are relegated to the supplementary materials.

**Theorem 1.** *For any breaking $G$ and any data $D$, there exists $\vec{\gamma} \in \bar{\Omega}$ such that $P_G(D) \cdot \vec{\gamma} = 0$.*

Theorem 1 implies that in Equation (1), $\inf_{\vec{\gamma} \in \Omega} g(D, \vec{\gamma})^T W_n g(D, \vec{\gamma})\} = 0$. Therefore, Step 2 can be replaced by: **2*** Let $\text{GMM}_G = \{\vec{\gamma} \in \Omega : P_G(D) \cdot \vec{\gamma} = 0\}$.

### 3.1 Uniqueness of Solution

It is possible that for some data $D$, $\text{GMM}_G(D)$ is empty or non-unique. Our next theorem characterizes conditions for $|\text{GMM}_G(D)| = 1$ and $|\text{GMM}_G(D)| \neq \emptyset$. A Markov chain (row stochastic matrix) $M$ is *irreducible*, if any state can be reached from any other state. That is, $M$ only has one communicating class.

**Theorem 2.** *Among the following three conditions, 1 and 2 are equivalent for any breaking $G$ and any data $D$. Moreover, conditions 1 and 2 are equivalent to condition 3 if and only if $G$ is connected.*

  *1. $(I + P_G(D)/m)^T$ is irreducible.*

  *2. $|GMM_G(D)| = 1$.*

  *3. $GMM_G(D) \neq \emptyset$.*

**Corollary 1.** *For the full breaking, adjacent breaking, and any top-$k$ breaking, the three statements in Theorem 2 are equivalent for any data $D$. For any position-$k$ (with $k \geq 2$) and any bottom-$k$ (with $k \leq m - 1$), 1 and 2 are not equivalent to 3 for some data $D$.*

Ford, Jr. [6] identified a necessary and sufficient condition on data $D$ for the MLE under PL to be unique, which is equivalent to condition 1 in Theorem 2. Therefore, we have the following corollary.

**Corollary 2.** *For the full breaking $G_F$, $|GMM_{G_F}(D)| = 1$ if and only if $|MLE_{PL}(D)| = 1$.*

### 3.2 Consistency

We say a breaking $G$ is *consistent* (for PL), if $\text{GMM}_G$ is consistent (for PL). Below, we show that some breakings defined in the last subsection are consistent. We start with general results.

**Theorem 3.** *A breaking $G$ is consistent if and only if $E_{d|\vec{\gamma}^*}[g(d, \vec{\gamma}^*)] = 0$, which is equivalent to the following equalities:*

$$\text{for all } i \neq j, \quad \frac{\Pr(c_i \succ c_j | \{\text{Pos}(c_i, d), \text{Pos}(c_j, d)\} \in G)}{\Pr(c_j \succ c_i | \{\text{Pos}(c_i), \text{Pos}(c_j)\} \in G)} = \frac{\gamma_i^*}{\gamma_j^*}. \tag{2}$$

**Theorem 4.** *Let $G_1, G_2$ be a pair of consistent breakings.*

  *1. If $G_1 \cap G_2 = \emptyset$, then $G_1 \cup G_2$ is also consistent.*

  *2. If $G_1 \subsetneq G_2$ and $(G_2 \setminus G_1) \neq \emptyset$, then $(G_2 \setminus G_1)$ is also consistent.*

Continuing, we show that position-$k$ breakings are consistent, then use this and Theorem 4 as building blocks to prove additional consistency results.

**Proposition 1.** *For any $k \geq 1$, the position-$k$ breaking $G_P^k$ is consistent.*

We recall that $G_T^k = \bigcup_{l=1}^{k} G_P^l$, $G_F = G_T^m$, and $G_B^k = G_F \setminus G_T^{m-k}$. Therefore, we have the following corollary.

**Corollary 3.** *The full breaking $G_F$ is consistent; for any $k$, $G_T^k$ is consistent, and for any $k \geq 2$, $G_B^k$ is consistent.*

**Theorem 5.** *Adjacent breaking $G_A$ is consistent if and only if all components in $\vec{\gamma}^*$ are the same.*

Lastly, the technique developed in this section can also provide an independent proof that the RC algorithm is consistent for BTL, which is implied by the main theorem in [16]:

**Corollary 4.** *[16] The RC algorithm is consistent for BTL.*

RC is equivalent to $GMM_{g_{RC}}$ that satisfies Condition 1. By checking similar conditions as we did in the proof of Theorem 3, we can prove that $GMM_{g_{RC}}$ is consistent for BTL.

The results in this section suggest that if we want to learn the parameters of PL, we should use consistent breakings, including full breaking, top-$k$ breakings, bottom-$k$ breakings, and position-$k$ breakings. The adjacent breaking seems quite natural, but it is not consistent, thus will not provide a good estimate to the parameters of PL. This will also be verified by experimental results in Section 4.

### 3.3 Complexity

We first characterize the computational complexity of our GMMs.

**Proposition 2.** *The computational complexity of the MM algorithm for PL [8] and our GMMs are listed below.*

- **MM:** $O(m^3 n)$ *per iteration.*

- **GMM (Algorithm 1) with full breaking:** $O(m^2 n + m^{2.376})$, *with* $O(m^2 n)$ *for breaking and* $O(m^{2.376})$ *for computing step* $2^*$ *in Algorithm 1 (matrix inversion).*

- **GMM with adjacent breaking:** $O(mn + m^{2.376})$, *with* $O(mn)$ *for breaking and* $O(m^{2.376})$ *for computing step* $2^*$ *in Algorithm 1.*

- **GMM with top-$k$ breaking:** $O((m + k)kn + m^{2.376})$, *with* $O((m + k)kn)$ *for breaking and* $O(m^{2.376})$ *for computing step* $2^*$ *in Algorithm 1.*

It follows that the asymptotic complexity of the GMM algorithms is better than for the classical MM algorithm. In particular, the GMM with adjacent breaking and top-$k$ breaking for constant $k$'s are the fastest. However, we recall that the GMM with adjacent breaking is not consistent, while the other algorithms are consistent. We would expect that as data size grows, the GMM with adjacent breaking will provide a relatively poor estimation to $\vec{\gamma}^*$ compared to the other methods.

Moreover in the statistical setting in order to gain consistency we need regimes that $m = o(n)$ and large $n$s are going to lead to major computational bottlenecks. All the above algorithms (MM and different GMMs) have linear complexity in $n$, hence, the coefficient for $n$ is essential in determining the tradeoffs between these methods. As it can be seen above the coefficient for $n$ is linear in $m$ for top-$k$ breaking and quadratic for full breaking while it is cubic in $m$ for the MM algorithm. This difference is illustrated through experiments in Figure 5.

Among GMMs with top-$k$ breakings, the larger the $k$ is, the more information we use in a single ranking, which comes at a higher computational cost. Therefore, it is natural to conjecture that for the same data, $GMM_{G_T^k}$ with large $k$ converges faster than $GMM_{G_T^k}$ with small $k$. In other words, we expect to see the following time-efficiency tradeoff among $GMM_{G_T^k}$ for different $k$'s, which is verified by the experimental results in the next section.

**Conjecture 1. (time-efficiency tradeoff)** *for any $k_1 < k_2$, $GMM_{G_T^{k_1}}$ runs faster, while $GMM_{G_T^{k_2}}$ provides a better estimate to the ground truth.*

## 4 Experiments

The running time and statistical efficiency of MM and our GMMs are examined for both synthetic data and a real-world sushi dataset [9]. The synthetic datasets are generated as follows.

- Generating the ground truth: for $m \leq 300$, the ground truth $\vec{\gamma}^*$ is generated from the Dirichlet distribution $\text{Dir}(\vec{1})$.

- Generating data: given a ground truth $\vec{\gamma}^*$, we generate up to 1000 full rankings from PL.

We implemented MM [8] for 1, 3, 10 iterations, as well as GMMs with full breaking, adjacent breaking, and top-$k$ breaking for all $k \leq m - 1$.

We focus on the following representative criteria. Let $\vec{\gamma}$ denote the output of the algorithm.

- *Mean Squared Error*: MSE $= E(\|\vec{\gamma} - \vec{\gamma}^*\|_2^2)$.

- *Kendall Rank Correlation Coefficient*: Let $K(\vec{\gamma}, \vec{\gamma}^*)$ denote the Kendall tau distance between the ranking over components in $\vec{\gamma}$ and the ranking over components in $\vec{\gamma}^*$. The Kendall correlation is $1 - 2\frac{K(\vec{\gamma}, \vec{\gamma}^*)}{m(m-1)/2}$.

All experiments are run on a 1.86 GHz Intel Core 2 Duo MacBook Air. The multiple repetitions for the statistical efficiency experiments in Figure 3 and experiments for sushi data in Figure 5 have been done using the odyssey cluster. All the codes are written in R project and they are available as a part of the package "StatRank".

## 4.1 Synthetic Data

In this subsection we focus on comparisons among MM, GMM-F (full breaking), and GMM-A (adjacent breaking). The running time is presented in Figure 2. We observe that GMM-A (adjacent breaking) is the fastest and MM is the slowest, even for one iteration.

The statistical efficiency is shown in Figure 3. We observe that in regard to the MSE criterion, GMM-F (full breaking) performs as well as MM for 10 iterations (which converges), and that these are both better than GMM-A (adjacent breaking). For the Kendall correlation criterion, GMM-F (full breaking) has the best performance and GMM-A (adjacent breaking) has the worst performance. Statistics are calculated over 1840 trials. In all cases except one, GMM-F (full breaking) outperforms MM which outperforms GMM-A (adjacent breaking) with statistical significance at 95% confidence. The only exception is between GMM-F (full breaking) and MM for Kendall correlation at $n = 1000$.

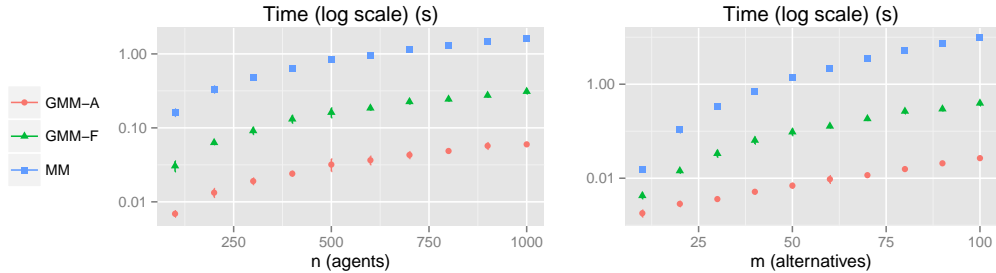

Figure 2: The running time of MM (one iteration), GMM-F (full breaking), and GMM-A (adjacent breaking), plotted in log-scale. On the left, $m$ is fixed at 10. On the right, $n$ is fixed at 10. 95% confidence intervals are too small to be seen. Times are calculated over 20 datasets.

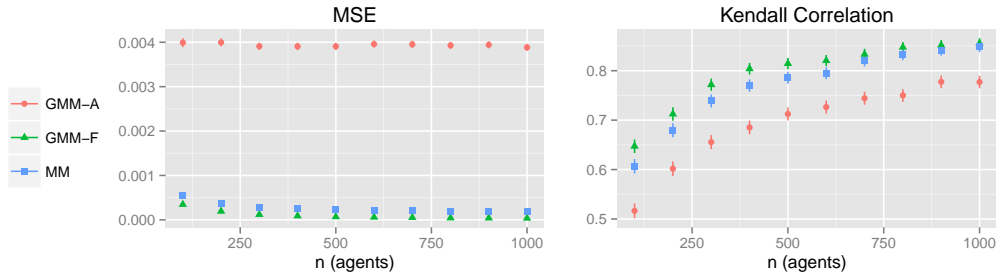

Figure 3: The MSE and Kendall correlation of MM (10 iterations), GMM-F (full breaking), and GMM-A (adjacent breaking). Error bars are 95% confidence intervals.

## 4.2 Time-Efficiency Tradeoff among Top-$k$ Breakings

Results on the running time and statistical efficiency for top-$k$ breakings are shown in Figure 4. We recall that top-1 is equivalent to position-1, and top-$(m-1)$ is equivalent to the full breaking.

For $n = 100$, MSE comparisons between successive top-$k$ breakings are statistically significant at 95% level from (top-1, top-2) to (top-6, top-7). The comparisons in running time are all significant at 95% confidence level. On average, we observe that top-$k$ breakings with smaller $k$ run faster, while top-$k$ breakings with larger $k$ have higher statistical efficiency in both MSE and Kendall correlation. This justifies Conjecture 1.

### 4.3 Experiments for Real Data

In the sushi dataset [9], there are 10 kinds of sushi ($m = 10$) and the amount of data $n$ is varied, randomly sampling with replacement. We set the ground truth to be the output of MM applied to all 5000 data points. For the running time, we observe the same as for the synthetic data: GMM (adjacent breaking) runs faster than GMM (full breaking), which runs faster than MM (The results on running time can be found in supplementary material B).

Comparisons for MSE and Kendall correlation are shown in Figure 5. In both figures, 95% confidence intervals are plotted but too small to be seen. Statistics are calculated over 1970 trials.

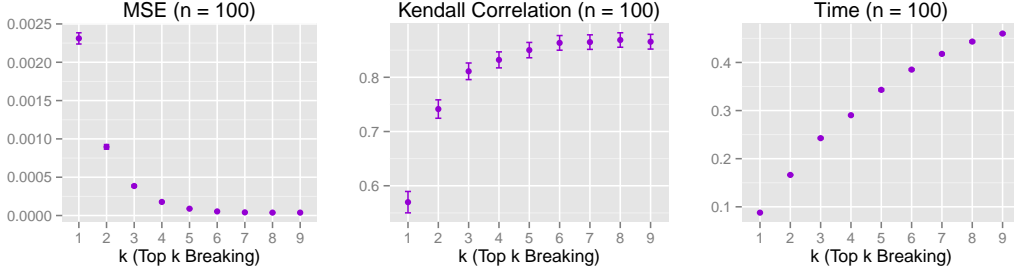

Figure 4: Comparison of GMM with top-$k$ breakings as $k$ is varied. The $x$-axis represents $k$ in the top-$k$ breaking. Error bars are 95% confidence intervals and $m = 10, n = 100$.

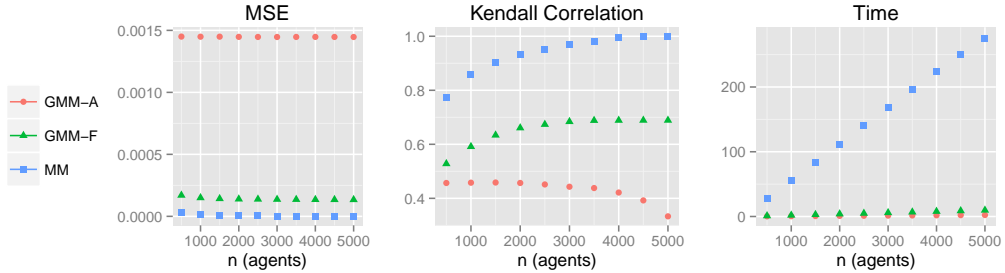

Figure 5: The MSE and Kendall correlation criteria and computation time for MM (10 iterations), GMM-F (full breaking), and GMM-A (adjacent breaking) on sushi data.

For MSE and Kendall correlation, we observe that MM converges fastest, followed by GMM (full breaking), which outperforms GMM (adjacent breaking) which does not converge. Differences between performances are all statistically significant with 95% confidence (with exception of Kendall correlation and both GMM methods for $n = 200$, where $p = 0.07$). This is different from comparisons for synthetic data (Figure 3). We believe that the main reason is because PL does not fit sushi data well, which is a fact recently observed by Azari et al. [1]. Therefore, we cannot expect that GMM converges to the output of MM on the sushi dataset, since the consistency results (Corollary 3) assumes that the data is generated under PL.

## 5 Future Work

We plan to work on the connection between consistent breakings and preference elicitation. For example, even though the theory in this paper is developed for full ranks, the notion of top-$k$ and bottom-$k$ breaking are implicitly allowing some partial order settings. More specifically, top-$k$ breaking can be achieved from partial orders that include full rankings for the top-$k$ alternatives.

### Acknowledgments

This work is supported in part by NSF Grants No. CCF- 0915016 and No. AF-1301976. Lirong Xia acknowledges NSF under Grant No. 1136996 to the Computing Research Association for the CIFellows project and an RPI startup fund. We thank Joseph K. Blitzstein, Edoardo M. Airoldi, Ryan P. Adams, Devavrat Shah, Yiling Chen, Gábor Cárdi and members of Harvard EconCS group for their comments on different aspects of this work. We thank anonymous NIPS-13 reviewers, for helpful comments and suggestions.

## Footnotes

[1] The BTL model in [16] is slightly different from that in this paper. Therefore, in this example we adopt an equivalent description of the RC algorithm.

[2]To simplify notation, we use $GMM_G$ instead of $GMM_{g_G}$.

[3]We need $k \geq 2$ since $G_B^k$ is empty.

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
