[Supplementary Material]

# Supplementary Materials for: Generalized Method-of-Moments for Rank Aggregation

## A proof

**Theorem 1.** For any breaking $G$ and any data $D$, there exists $\vec{\gamma} \in \bar{\Omega}$ such that $P_G(D) \cdot \vec{\gamma} = 0$.

*Proof.* For any $G$ and $D$, $(I + P_G(D)/m)^T$ is a row stochastic matrix, which means that the corresponding Markov chain has a stationary distribution $\vec{\gamma}$. It follows that $P_G(D) \cdot \vec{\gamma} = 0$ and $\vec{\gamma} \in \bar{\Omega}$. $\qquad\square$

**Theorem 2.** Among the following three conditions, 1 and 2 are equivalent for any breaking $G$ and any data $D$. Moreover, conditions 1 and 2 are equivalent to condition 3 if and only if $G$ is connected.

1. $(I + P_G(D)/m)^T$ is irreducible.
2. $|\text{GMM}_G(D)| = 1$.
3. $\text{GMM}_G(D) \neq \emptyset$.

*Proof.* We first prove that 1 and 2 are equivalent.

$1 \Rightarrow 2$: By Proposition 1.14 in [10], we have that if $(I + P_G(D)/m)^T$ is irreducible, then $(I + P_G(D)/m)^T$ has a unique positive stationary distribution, which means that $|\text{GMM}_G(D)| = 1$.

$2 \Rightarrow 1$: suppose on the contrary that $(I + P_G(D)/m)^T$ is not irreducible. There are two cases.

Case 1: there exists an inessential state. Then, for any stationary distribution, the inessential state must have 0 probability (Proposition 1.25 in [10]). This means that $\text{GMM}_G(D) = \emptyset$.

Case 2: there is no inessential state. In this case all essential communicating classes do not communicate. Therefore, any convex combination of their respective stationary distributions is an overall stationary distribution. This means that $|\text{GMM}_G(D)| = \infty$.

We next prove that 1 and 3 are equivalent for any $D$ if and only if $G$ is connected. Notice that the only possibility for 1 and 3 to be not equivalent is Case 2 above.

The "if" part: if $G$ is connected, then Case 2 is not possible.

The "only if" part: if $G$ is not connected, without loss of generality let $\{1, \ldots, m\} = M_1 \cup M_2$ such that $M_1 \cap M_2 = \emptyset$. W.l.o.g. let $M_1 = \{1, \ldots, m'\}$ and $M_2 = \{m' + 1, \ldots, m\}$. Let $D = \{[R_1 \succ R_2]\}$, where $R_1$ is any ranking over $\{c_1, \ldots, c_{m'}\}$ and $R_2$ is any ranking over $\{c_{m'+1}, \ldots, c_m\}$. Therefore, there is a positive stationary probability for $M_1$ and a positive stationary probability for $M_2$. Any convex combination of these two stationary probabilities is a positive stationary probability for $P_G(D)$. $\qquad\square$

**Theorem 3.** A breaking $G$ is consistent if and only if $E_{d|\vec{\gamma}^*}[g(d, \vec{\gamma}^*)] = 0$, which is equivalent to the following equalities:

$$\text{for any } i \neq j, \quad \frac{\Pr(c_i \succ c_j | \{c_i, c_j\} \in G)}{\Pr(c_j \succ c_i | \{c_i, c_j\} \in G)} = \frac{\gamma_i}{\gamma_j}. \tag{3}$$

*Proof.* We first prove the following lemma.

**Lemma 1.** *For any breaking $G$, $\vec{\gamma}^* \in \Omega$, $E_{d|\vec{\gamma}^*}[P(d)] \cdot \vec{\gamma} = 0$ has a unique solution in $\Omega$.*

**Proof sketch:** It is not hard to verify that $I + E_{d|\vec{\gamma}^*}[P(d)]/m$ is a column stochastic matrix whose entries are all strictly positive. So the stationary distribution $\vec{\gamma}'$ of the Markov Chain with transition

matrix $(I + E_{d|\vec{\gamma}^*}[P(d)]/m)^T$ is unique and strictly positive. It follows that $\vec{\gamma}' \in \Omega$ is the solution to $E_{d|\vec{\gamma}^*}[P(d)] \cdot \vec{\gamma} = 0$. □

For the "if" part, we apply Theorem 2.2 in [7]. By Lemma 1 and the premise of the theorem, Condition 1 is satisfied. To show that $G$ is consistent, it suffice to prove that $\text{GMM}_G$ satisfies Assumption 2.1 to 2.6 and the three premises in the statement of Theorem 2.2 in [7]. We slightly abuse the notation by adding one component to $g_G$: the $(m+1)$th component is $\vec{1} \times \vec{\gamma} - 1$.

Assumption 2.1: $D$ is stationary and ergodic. This holds because in PL, data in $D$ are generated i.i.d.

Assumption 2.2: $\Omega$ is a separable metric space. Since $\mathbb{R}^m$ is separable and $\Omega$ is an open subset of $\mathbb{R}^m$, $\Omega$ is also separable.

Assumption 2.3: $g_G(\cdot, \vec{\gamma})$ is Borel measurable for each $\vec{\gamma} \in \Omega$ and $g_G(d, \cdot)$ is continuous on $\Omega$ for each $d$. Since the domain of $g_G(\cdot, \vec{\gamma})$ discrete, $g_G(\cdot, \vec{\gamma})$ is continues, which means that $g_G(\cdot, \vec{\gamma})$ is Borel measurable. We note that $g_G(d, \cdot)$ is linear, which means that it is continuous.

Assumption 2.4: $E_{d|\vec{\gamma}^*}[g_G(d, \vec{\gamma})]$ exists and is finite for all $\vec{\gamma} \in \Omega$, and $E_{d|\vec{\gamma}^*}[g_G(d, \vec{\gamma}^*)] = 0$. The former is because $E_{d|\vec{\gamma}^*}[g_G(d, \vec{\gamma})]$ is linear in $\vec{\gamma}$ and $\Omega$ is bounded. The latter is the assumption.

Assumption 2.5: The sequence $\mathcal{W}$ converges almost surely to a positive semi-definite matrix. This holds since $W_n = I$ for all $t$.

Assumption 2.6 is satisfied by the definition of $\text{GMM}_G$.

Premise (1): Since $\mathbb{R}^{m+1}$ is locally compact and $\Omega$ is an open subset of $\mathbb{R}^{m+1}$, $\Omega$ is also locally compact.

Premise (2) and (3). Since $\begin{bmatrix} E[P_G(d)] \\ \vec{1} \end{bmatrix}$ is full rank, following the discussion after Theorem 2.2 in [7], we have that (2) and (3) must be satisfied as well.

For the "only if" part, we need to show that there exists a neighborhood $\mathcal{N}$ of $\vec{\gamma}^*$ such that for any $\vec{\gamma} \in \mathcal{N}$, there exists $n^*$ such that for any $n \geq n^*$, $g_G(D, \vec{\gamma}) \neq 0$ with high probability. We note that as $n \to \infty$, $P_G(D) \to E_{P_G(d)}$, which means that $P_G(D) \cdot \vec{\gamma} \neq 0$ with high probability for a sufficiently small neighborhood of $\vec{\gamma}^*$ and sufficient large dataset.

We next show that $E_{d|\vec{\gamma}^*}[g(d, \vec{\gamma}^*)] = 0$ is equivalent to Equation (3). By Lemma 1, $\vec{\gamma}^*$ is the only nonzero solution to $E_{d|\vec{\gamma}^*}[g(d, \vec{\gamma})] = 0$. Also $\vec{\gamma}^*$ is the only solution to Equation (3). This means that they are equivalent. □

**Theorem 4.** Let $G_1, G_2$ be a pair of consistent breakings.

1. If $G_1 \cap G_2 = \emptyset$, then $G_1 \cup G_2$ is also consistent.
2. If $G_1 \subsetneq G_2$, then $G_2 \setminus G_1$ is also consistent.

*Proof.* We first prove 1. By Theorem 3, for $i \neq j$ we have:

$$\frac{\Pr(c_i \succ c_j | \{c_i, c_j\} \in G_1)}{\Pr(c_j \succ c_i | \{c_i, c_j\} \in G_1)} = \frac{\gamma_i}{\gamma_j}$$

$$\frac{\Pr(c_i \succ c_j | \{c_i, c_j\} \in G_2)}{\Pr(c_j \succ c_i | \{c_i, c_j\} \in G_2)} = \frac{\gamma_i}{\gamma_j}$$

Then we have:

$$\frac{\Pr(c_i \succ c_j | \{c_i, c_j\} \in G_1 \cup G_2)}{\Pr(c_j \succ c_i | \{c_i, c_j\} \in G_1 \cup G_2)} = \frac{\Pr(c_i \succ c_j | \{c_i, c_j\} \in G_1) \Pr(G_1) + \Pr(c_i \succ c_j | \{c_i, c_j\} \in G_2) \Pr(G_2)}{\Pr(c_j \succ c_i | \{c_i, c_j\} \in G_1) \Pr(G_1) + \Pr(c_j \succ c_i | \{c_i, c_j\} \in G_2) \Pr(G_2)} = \frac{\gamma_i}{\gamma_j}$$

Where $\Pr(G) = \Pr(\{c_i, c_j\} \in G)$ is the probability that $\{c_i, c_j\} \in G$. This shows the consistency of $G_1 \cup G_2$. The proof of 2 is similar. □

**Proposition 1.** For any $k \geq 1$, the position-$k$ breaking $G_P^k$ is consistent.

*Proof.* Define $T_k$ the set of top $k$ alternatives in a ranking. And $\pi(k)$ is the $k$th ranked alternative. Then we have:

$$\frac{\Pr(c_i \succ c_j | \{c_i, c_j\} \in G_k)}{\Pr(c_j \succ c_i | \{c_i, c_j\} \in G_k)} = \frac{\Pr(\pi(k) = c_i | c_i, c_j \notin T_{k-1})}{\Pr(\pi(k) = c_j | c_i, c_j \notin T_{k-1})} = \frac{\sum_{T_{k-1}} \Pr(\pi(k) = c_i | T_{k-1}) \Pr(T_{k-1} | c_i, c_j \notin T_{k-1})}{\sum_{T_{k-1}} \Pr(\pi(k) = c_j | T_{k-1}) \Pr(T_{k-1} | c_i, c_j \notin T_{k-1})}$$

And since we are conditioning on $T_{k-1}$ and we know $c_i, c_j \notin T_{k-1}$ , using Luce's IIA, we have:

$$\frac{\Pr(\pi(k) = c_i | T_{k-1})}{\Pr(\pi(k) = c_j | T_{k-1})} = \frac{\frac{\gamma_i}{\sum_{l \notin T_{k-1}} \gamma_l}}{\frac{\gamma_j}{\sum_{l \notin T_{k-1}} \gamma_l}} = \frac{\gamma_i}{\gamma_j}$$

Hence:

$$\frac{\Pr(c_i \succ c_j | \{c_i, c_j\} \in G_k)}{\Pr(c_j \succ c_i | \{c_i, c_j\} \in G_k)} = \frac{\gamma_i}{\gamma_j}$$

and this concludes the proof. □

**Theorem 5.** Adjacent breaking $G_A$ is consistent if and only if all components in $\vec{\gamma}^*$ are the same.

*Proof.* It is not hard to check that if all the $\gamma_i$'s are the same, then adjacent breaking is consistent. We next show the "only if" part.

Suppose $\gamma_j > \gamma_i$. We know that:

$$\frac{\Pr(c_i \succ c_j)}{\Pr(c_j \succ c_i)} = \frac{\gamma_i}{\gamma_j}$$

We next show that for the adjacent breaking we have:

$$\frac{\Pr(c_i \succ c_j | \{c_i, c_j\} \in G_A)}{\Pr(c_j \succ c_i | \{c_i, c_j\} \in G_A)} > \frac{\gamma_i}{\gamma_j} \tag{4}$$

To show the above inequality we will condition on the appearance of $c_i, c_j$ as the $k$th pair $(G_{A_k})$.

$$\frac{\Pr(c_i \succ c_j | \{c_i, c_j\} \in G_A)}{\Pr(c_j \succ c_i | \{c_i, c_j\} \in G_A)} = \frac{\sum_k \Pr(c_i \succ c_j | \{c_i, c_j\} \in G_{A_k}) \Pr(G_{A_k})}{\sum_k \Pr(c_j \succ c_i | \{c_i, c_j\} \in G_{A_k}) \Pr(G_{A_k})}$$

It suffices to show:

$$\frac{\Pr(c_i \succ c_j | \{c_i, c_j\} \in G_{A_k})}{\Pr(c_j \succ c_i | \{c_i, c_j\} \in G_{A_k}))} > \frac{\gamma_i}{\gamma_j}$$

Again we can condition on the $T_{k-1}$, the set of alternatives ranked 1 to $k - 1$.

$$\frac{\Pr(c_i \succ c_j | \{c_i, c_j\} \in G_{A_k})}{\Pr(c_j \succ c_i | \{c_i, c_j\} \in G_{A_k})} = \frac{\sum_{T_{k-1}} \Pr(c_i \succ c_j | \{c_i, c_j\} \in G_{A_k}, T_{k-1}) \Pr(T_{k-1})}{\sum_{T_{k-1}} \Pr(c_j \succ c_i | \{c_i, c_j\} \in G_{A_k}, T_{k-1}) \Pr(T_{k-1})}$$

Define, $\gamma_T = 1 - \gamma_i - \gamma_j - \sum_{l \in T_{k-1}} \gamma_l$, then using the assumption $\gamma_i < \gamma_j$, we have:

$$\frac{\Pr(c_i \succ c_j | \{c_i, c_j\} \in G_{A_k}, T_{k-1})}{\Pr(c_j \succ c_i | \{c_i, c_j\} \in G_{A_k}, T_{k-1})} = \frac{\gamma_T + \gamma_i}{\gamma_T + \gamma_j} > \frac{\gamma_i}{\gamma_j}$$

This shows that inequality (4) holds. □

# B   More Experimental Results

Figure 6: The running time for MM (10 iterations), GMM (full breaking), and GMM (adjacent breaking).