[Reviews · NeurIPS 2013]

Submitted by Assigned_Reviewer_1

The paper investigates a technique for efficiently combining multiple rankings into a Placket Luce model. The key idea is to use so-called breakings, which are subsets of the total ranking information. The authors show that consistency is an important property, which is, e.g., satisfied by top-k rankings (1,2 > 3,4,5), but not by an adjanceny ranking (1 > 2,2 > 3,3 > 4,4 > 5).

The paper is o.k. The presentation of the method appears occassionally overly formal, but the examples make the message quite clear.

I did not understand why the method is called "Generalized Method of Moments". What is the method of moments, and how has it been generalized here? For a while I thought that the MM in the experimental comparison stands for the non-generalized MM, but it stands for "Minorize Maximization".

The authors also assume that the reader is familiar with the MM method (which I am not). Thus, it is difficult to appreciate the contribution. The authors claim that MM is standard method for rank aggregation, but I think they should have also considered other methods.

Why are the top-k and the bottom-k breaking not symmetric in structure? Why would anyone use the bottom-k breaking?


Summary: Overall, the paper appears quite reasonable, but I don't think that this contribution will make a strong impact. In any case, the experimental evidence (improvement over MM on synthetic datasets and on the Sushi datasets) does not strike me as convincing enough.

Submitted by Assigned_Reviewer_5

This paper proposes a class of algorithms for rank aggregation, referred to as GMM algorithms. The key idea of the algorithms is to employ GMM for rank aggregation and to break the full rankings in rank aggregation into pairwise comparisons. The paper gives conditions for the uniqueness and consistency of GMM algorithms. It also shows by theory and experiments that the proposed algorithms run faster than the baseline algorithm of MM, while achieving competitive statistical efficiency.

The paper is very well written, and it is easy to understand the major points of the paper. The proposed methods appear to be sound, and theoretical and empirical studies on the algorithms are also convincing. The work is novel and represents significant contribution to the research on learning to rank.

Minor issues:
* Page 4, line 173, position-k breaking G_P^k is only defined for k \ge 2. However, in line 200, G_P^1 is given.
* Page 3, line 152, it is not clear whether Definition 1 is about breaking, or about the GMM method GMM_G(D).
Summary: This paper is well written, and the work is sound, novel, and significant. I vote for accept for the paper.

Submitted by Assigned_Reviewer_6

The authors propose a generalized method of moments (GMM) approach for computing parameters of the Plackett-Luce model. The method first 'breaks' the complete rankings into pairwise comparisons which are then used to estimate the necessary quantities at each iteration of the algorithm.

I find the proposed approach to be interesting, I especially like the analysis of the relationship between the breaking type and consistency/quality of found solutions which leads to a trade-off between time and efficiency.

The major drawback of the GMM methods is its applicability. All results in the paper hold for full rankings only while in practice most preference aggregation problems typically have partial rankings/preferences. The authors mention extension to the partial case in future work but I think having at least some results on that would significantly increase the impact of the paper.

I also have some comments regarding the experiments which are summarized below.

-The real data experiment is not very convincing. It seems strange to compare models on a dataset where neither model provides a good fit, especially since solution from one of these models is used as ground truth. Why was this data chosen?

-I would want to see a comparison with a gradient descent procedure on the PL log likelihood. In my experience gradient descent works well on PL models especially in settings where relative order of \gamma's is more important than the absolute magnitude (i.e. Kendall correlation criteria). Careful implementation should also be more efficient than GMM per iteration and require less storage. Do you have any results on this?
Summary: I find the proposed approach to be interesting and promising but in the current state it will have limited impact on the relevant research area.

Submitted by Assigned_Reviewer_7

This paper proposes a generalized method of moments approach to rank aggregation, specifically by estimating the parameters of a Plackett-Luce (PL) model from a sample of rankings. The proposed method works by decomposing rankings into pairwise comparisons, which are then used to build a transition matrix, the stationary distribution of which is used to parameterize the PL model. The authors establish consistency results for different ways of "breaking" the input rankings into pairwise comparisons, and experimentally validate the efficiency of the approach on two datasets.

Overall, the paper is clearly written, and the method seems sound. The "breaking" technique is both interesting and intuitive.

My two main concerns with this paper are the practical applicability of the method, and the thoroughness of experimental evaluation.

In both the analysis and the experiments, the authors focus on a small-m/large-n regime, where a large number of rankings are given over a small set of items. While consistency is generally a desirable property, I'm not convinced that it's the main quality of interest for rank aggregation. Many of the practical applications mentioned by the authors in the introduction (e.g., meta-search) are large-m/small-n settings in which relatively few rankings are provided over a large set of items. The authors do discuss the computational complexity of inference, but there is no discussion or evaluation of accuracy in this regime, and it's not clear how well the proposed method would perform.

The experiments on synthetic and real data do illustrate qualitative differences between different breaking strategies, both for speed and accuracy (RMSE and Kendall correlation). However, neither metric seems specifically appropriate for top-k breakings (figure 4), as the scores may be polluted by errors low in the ranking which have no qualitative effect in practice. It would be helpful to see at least one position-dependent score here.


Minor comments:

Line 83: \gamma* is not used in the definition of Pr_PL. Should this be \gamma?

Line 95: should the norm ||a||_W be squared here? Is this intended as a proper norm (i.e., W positive definite), or is it just meant as a convenient notation? Is it necessary to introduce the W notation here anyway, since it seems that the remainder of the paper sets W=I?

Line 102: GMM_g(D,W) is set-valued, should converge to the set {\gamma*}, not the point \gamma*.

Line 124: section 3 title misspells "Plackett"

Line 245, Eq. 2: should both the numerator and denominator here be in terms of Pos(c, d)? Or Pos(c)?

Line 377: broken reference in footnote 4
Summary: This paper proposes a generalized method of moments approach to estimating the parameter vector for a Plackett-Luce ranking model. The paper is clearly written and the analysis is interesting, but it's not entirely clear how practically applicable the method is, and the experimental evaluation could be more thorough.
Author Feedback

Author rebuttal: We thank all the reviewers for their insightful comments.
R: Reviewers’ Comment
A: Authors’ response

Reviewer 1:

R: The paper investigates a technique for efficiently combining multiple rankings…

R: The paper is o.k … quite clear.

R: I did not understand why the method is called "Generalized Method of Moments". What is the method of moments, and how has it been generalized here?

A: We will add a brief explanation and pointers: Method of moments is a statistical estimation method. The idea is to write equations using the true moments of the model as a function of parameters and the empirical moments which can be computed form the data

R: For a while ... but it stands for "Minorize Maximization".

A: Yes, we also thought about this confusion, MM for "Minorize Maximization" can be confused by method of moments, however, Method of Moments is generally abbreviated to MoM in statistics literature (while MM has become standard for the former).

R: The authors also … I think they should have also considered other methods.

A: Thanks. Other methods are all iteration based (and we will also add a pointer that explains a connection between EM and the more widely known E-M algorithm). Still, the main contribution here is not devising GMM, but rather, applying breaking to have a fast computation of parameters.


R: Why are the top-k and the bottom-k breaking not symmetric in structure?

A: We construct bottom-k by complementing top-k and using Theorem 4. A bottom breaking definition that is symmetric with top-k wouldn't be consistent.

R: Why would anyone use the bottom-k breaking?

A: It has a computational advantage. Similar to the discussion in section 3.3, bottom k has a sub-linear complexity for breaking in terms of the number of alternatives, and is consistent due to corollary 3. And, using theorem 4, bottom-k breaking can be used as a building block for more complicated breakings.

A: Moreover, in applications such as online games, where we want to rank players, if the weakest players leave the game first due to game dynamics then bottom-k information is available before top-k information.

Reviewer 2:
R: This paper proposes … competitive statistical efficiency.

R: The paper is very well written…represents significant contribution to the research on learning to rank.

R: Minor issues: Page4…

A: Thanks. We will be sure to fix these issues.

Reviewer 3:

R: The authors propose…

R: I find…

R: The major … The authors mention extension to the partial case in future work but … impact of the paper.

A: This is an interesting point. Even though the theory is developed for full ranks, the notion of top-k and bottom-k breaking are implicitly allowing some partial order settings. For example top-3 breaking can be achieved from partial orders that include full rankings for the top-3. We will add a comment on this into the paper.


R: I also … regarding the experiments … below.

R: -The real data … Why was this data chosen?

A: We think the experiment shows an improvement over MM using breaking, both in regard to run time and statistic efficiency, complementing the theoretical results. The fit is better than on a voting data set we have, and we don’t know of other publicly available data where we’d expect P-L to be a better fit.

R: -I would want to … Careful implementation should also be more efficient than GMM per iteration and require less storage. Do you have any results on this?

A: Thanks for raising this point and yes, it would also be interesting to explore gradient descent methods. Most of papers we read for parameter estimation of PL used MM, which is why we adopt MM as the standard for comparison. [We assume the reviewer meant “more efficient than MM per iteration".]

Reviewer 4:

R: This paper proposes a generalized … transition matrix, the stationary distribution of which is used to parametrize the PL model.

A: Note: our approach is a bit different, we take a method of moments approach rather than finding the stationary distribution using Markov chain approaches.


R: The authors establish...

R: Overall, the paper is clearly written...

R: My two main concerns …

R: In both … few rankings are provided over a large set of items.

A: This is a very reasonable comment, however, most of the methods for rank aggregation using statistics considers big n/small m (e.g. Negahban et al. NIPS 2012). This is mainly due to the consistency property of the estimators which leads to unbiasedness (this is also highlighted in Negahban et al.)

A: We agree that the method should be examined for other environments, and we will look for a suitable data set in the extended version of this paper.


R: The authors do discuss…it's not clear how well the proposed method would perform.

A: We have evaluated the accuracy (i.e., statistical efficiency) through experiments in Section 4. For example figure 4 shows the time and accuracy (in terms of MSE and Kendall correlation) trade off.

R: The experiments on synthetic and real data do illustrate … (RMSE and Kendall correlation). However, neither metric … effect in practice. It would be helpful to see at least one position-dependent score here.


A: Yes, this is a good point. We will include one or two position-dependent metrics as well.


R Minor comments: Line 83: \gamma* I …_PL. Should this be \gamma?
A: yes


R: Line 95: should the norm ||a||_W be squared here?
A: yes

R: Is … a proper norm (i.e., W positive definite), or … convenient notation?
A: W should be positive definite.

R: Is it necessary … the remainder of the paper sets W=I?
A: No, we kept the W notation to be consistent with the original GMM definition.

R: Line 102: GMM_g(D,W) is … not the point \gamma*.

R: Line 124: section 3 …

R: Line 245, Eq. 2: should … Pos(c, d)? Or Pos(c)?

A: Pos(c, d)

R: Line 377: broken…

A: Thanks for catching these subtle points.